# Germination-Induced Enhancement of Brown Rice Noodle Nutritional Profile and Gut Microbiota Modulation

**DOI:** 10.3390/foods13142279

**Published:** 2024-07-19

**Authors:** Ruiyun Chen, Huibin Zhang, Jiamei Cai, Mingxi Cai, Taotao Dai, Yunfei Liu, Jianyong Wu

**Affiliations:** 1State Key Laboratory of Food Science and Resources, Nanchang University, Nanchang 330047, China; 2Jiangxi General Institute of Testing and Certification, Nanchang 330052, China; 3Institute of Applied Chemistry, Jiangxi Academy of Sciences, Nanchang 330096, China

**Keywords:** germination, brown rice, noodles, digestibility, *in vitro* fermentation

## Abstract

This study explored how germination influences the starch digestion and intestinal fermentation characteristics of brown rice noodle. The study began with *in vitro* starch digestion tests to assess how germination affects starch digestibility in brown rice noodles, revealing an increase in rapidly digestible starch content and a decrease in resistant starch content. Subsequently, an *in vitro* human fecal fermentation model was used to simulate the human intestinal environment, showing that germination altered pH levels and the production of short-chain fatty acids, particularly by increasing propionate while decreasing acetate and butyrate. Additionally, the study noted a decrease in gut microbiota diversity following fermentation, accompanied by an increase in *Megamonas* growth and a decrease in *Bacteroides* and *Bifidobacterium*. In conclusion, these findings suggest that germination could enhance the nutritional value and intestinal probiotic properties of brown rice noodles. This research contributes valuable insights into the role of germination in improving the nutritional properties of rice-based products and provides a foundation for further exploration into the development of health-promoting rice noodles.

## 1. Introduction

Rice noodles, a dietary staple in Southern China and Southeast Asia, are made from polished rice and brown rice using processes that include washing, soaking, crushing, fermenting, gelatinizing, extruding, and drying, resulting in a range of shapes [1]. Compared to polished rice noodles, brown rice noodles, which are rich in bran fiber and polyphenols, are gaining popularity among consumers. However, the presence of the cortex makes the flavor of brown rice noodle less palatable.

Germination, an economical technique, enhances the quality of brown rice noodles [2]. During germination, seeds undergo metabolic changes, releasing plant hormones like gibberellins, abscisic acid, and ethylene [3]. These hormones trigger enzymes such as amylase, lipase, and protease, which are crucial for breaking down nutrients and restructuring the seed’s components. This enzymatic activity mobilizes stored nutrients, facilitating growth. Germination also boosts bioactive compounds like polyphenols (tocopherols, tocotrienols) and γ-aminobutyric acid, enhancing nutritional value [4]. γ-Aminobutyric acid, a neurotransmitter, offers benefits like reducing anxiety and blood pressure; it is typically low in brown rice but increases throughout the germination period [5,6,7,8]. Therefore, germination confers distinctive characteristics to grains, enhancing their utility in nutritional value, thereby positioning them as beneficial functional foods.

Germination ameliorates the cooking quality, texture, sensory attributes, and nutritional profile of brown rice noodle, as demonstrated in our previous research [9]. Meanwhile, the content of starch, dietary fiber, γ-Aminobutyric acid, and polyphenols varied during the germination of brown rice. These nutrients have a positive effect on regulating intestinal microbiota. During the digestion process, 90% to 95% of dietary fiber (including resistant starch) and polyphenol polymers resist enzymatic digestion in the upper intestine and pass through to the colon [10]. Subsequently, undigested dietary polyphenols, resistant starch, and dietary fiber are fermented by gut microbiota into absorbable small molecular phenolic metabolites and short-chain fatty acids (SCFAs), such as acetic acid, propionic acid, and butyric acid, which are beneficial to human health [11,12,13]. Moreover, γ-Aminobutyric acid has been reported to improve high-fat diet-induced dysbiosis of gut microbiota [14]. Nevertheless, the impact of germinated brown rice noodle on gut microbiota remains unclear. Changes in dietary fiber and polyphenols due to germination in brown rice noodles may influence intestinal fermentation characteristics and microbiota composition.

In this study, we hypothesized that germination alters the nutritional and intestinal probiotic properties of brown rice noodles, a topic not extensively reported. Thus, we prepared both brown rice noodles and germinated brown rice noodles to investigate their starch digestion characteristics and intestinal probiotic properties using an in vitro human fecal fermentation model.

## 2. Materials and Methods

### 2.1. Materials

Brown rice (younian) was purchased from the local supermarket (Nanchang, Jiangxi province, China). Human salivary α-amylase, neutral protease, and standard dialysis bags were sourced from Shanghai Yuan Ye Biotechnology Co., Ltd. (Shanghai, China). Pepsin was purchased from Shanghai Aladdin Biochemical Technology Co., Ltd. (Shanghai, China). and short-chain fatty acid standards were obtained from Xiyachem Technology Co., Ltd. (Linyi, China). All other chemicals and reagents utilized in this research maintained analytical-grade purity.

### 2.2. Preparation of Brown Rice Noodles and Germinated Brown Rice Noodles

Brown rice germination was conducted according to the method described by Cáceres et al. [15]. The brown rice was washed with distilled water, surface-disinfected with 0.1% sodium hypochlorite for 30 min, and subsequently rinsed with distilled water until it reached a neutral pH. Then, the brown rice was soaked in distilled water at 28 °C for 12 h, drained, and placed on a germination tray covered with moistened germination paper. The germination tray containing the brown rice was placed in a constant temperature and humidity incubator set to 28 °C and 90% humidity for 18 h to obtain germinated brown rice. Next, the brown rice or germinated brown rice was ground used a grinding machine, then the moisture was removed by centrifugation and loosened to pass through a 40-mesh sieve to obtain the flour. The flour was then fed into the feeding device of a twin-screw extruder (FMHE36-24, Fumak Food Engineering Technology Co., Ltd., Changsha, China), using a die with a diameter of 2.2 mm. The extrusion parameters are set as follows: feed rate is 22.5 kg/h, screw speed is 100 rpm, moisture content is 8%, and the temperatures of the five sections of the twin-screw extruder were all set at 105 °C. The extruded noodles were placed in incubator at 40 °C and 80% humidity for aging for 8 h. Subsequently, they were air-dried at room temperature and sealed in self-sealing bags. The brown rice noodles and germinated brown rice noodles were named BRN and GBRN, respectively.

### 2.3. In Vitro Digestibility of Starch in Noodles

The in vitro digestibility of starch in the noodles was determined according to previously described methods with slight modifications [16]. To simulate the size of noodles after chewing, the cooked noodles were freeze-dried and crushed into particles of 20–30 mesh. Next, 200 mg of noodles particles was suspended in 20 mL CH_3_COONa solution (pH 5.2, 0.2 mol/L) and then stirred magnetically at 37 °C and 100 rpm for 10 min. Then, 200 μL of mixed enzyme solution (396 U/mL amyloglucosidase and 360 U/mL α-amylase) was added, and digestion was conducted in a magnetic stirring water bath at 37 °C and 100 rpm. Samples of enzymatic hydrolysate were taken at each time point (0, 20, 40, 60, 90, 120, and 180 min) and the reaction was terminated with anhydrous ethanol. Finally, the glucose content was determined by the GODPOD kit method (Megazyme glucose assay kit). The amount of glucose produced was measured at 510 nm using a microplate reader (SpectraMax M2, Molecular Devices, San Jose, CA, USA). The starch hydrolysis rate (%) was calculated using the following formula (Equation (1)).
(1)$$Starch\;hydrolysis\;rate\;\left(\%\right)=\frac{S_{h}}{TS}=\frac{0.9\times G_{h}}{TS}\times 100$$
where *S_h_* stands for hydrolyzed starch of noodles. *TS* stands for total starch of origin noodles. *G_h_* stands for the glucose content after starch was hydrolyzed. The glucose content was converted into the percentage of digested starch by multiplying it by 0.9.

Subsequently, starch fraction contents were calculated using the following formulas (Equations (2)–(4)).
(2)$$RDS\;\left(\%\right)=((G_{20}-G_{0})\times 0.9÷TS)\times 100$$
(3)$$SDS\;\left(\%\right)=((G_{120}-G_{20})\times 0.9÷TS)\times 100$$
(4)$$RS\;\left(\%\right)=((TS-(RDS+SDS))÷TS)\times 100$$

Here, *RDS*, *SDS*, and *RS* denote rapidly digestible starch, slowly digestible starch, and resistant starch, respectively. *G*_0_, *G*_20_, and *G*_120_ represent the glucose content levels in the digesta at 0, 20, and 120 min, respectively. Glucose content was converted to the percentage of digested starch by multiplying by a factor of 0.9.

Next, the first-order kinetics were analyzed by fitting the digestibility curves to the integrated equation (Equation (5)).
(5)$$C_{t}=C_{\infty }\times (1-e^{-kt})$$

Here, *t* represents the sampling time in minutes, *k* (min^−1^) is the starch digestion rate coefficient, *C_t_* denotes the amount of digested starch at time *t*, and *C_∞_* represents the amount of digested starch at the end of digestion.

### 2.4. In Vitro Fermentation Characteristics of Noodles

#### 2.4.1. In Vitro Upper Gastrointestinal Tract Digestion 

The samples underwent upper gastrointestinal tract digestion following the method outlined and the electrolyte stock solution formula for oral (pH = 7.0), gastric (pH = 3.0) and intestinal (pH = 7.0) fluid electrolyte stock solutions as described by Minekus et al. [17]. Initially, the noodles were cooked under optimal cooking conditions (in which the white core of the rice noodles disappeared completely), freeze-dried, and then crushed into particles of 20–30 mesh size (simulating the size of the vermicelli after chewing). Next, 5 g of noodle sample (dry weight) was suspended in 3.5 mL oral electrolyte stock solution formula and combined with 0.5 mL of α-amylase solution (1500 U/mL), 25 μL of CaCl_2_(H_2_O)_2_ (0.3 mol/L), and 9.75 μL of distilled water. The mixture was stirred continuously at 37 °C and 100 rpm for 2 min. Subsequently, 7.5 mL of gastric fluid electrolyte stock solution, 1.6 mL of pepsin (2500 U/mL), 25 μL of CaCl_2_(H_2_O)_2_ (0.3 mol/L), 0.2 mL of HCl (1 mol/L), and 9.75 μL of distilled water were added. The mixture was stirred continuously at 37 °C, 100 rpm for 2 h. Following this, 11 mL of intestinal fluid electrolyte stock solution formula, 5 mL of trypsin (80 0 U/mL), 40 μL of CaCl_2_(H_2_O)_2_ (0.3 mol/L), 2.5 mL of ox bile salt (160 mmol/L), 0.15 mL of NaOH (1 mol/L), and 1.31 mL of distilled water were added. The mixture was stirred continuously at 37 °C, 100 rpm for 2 h. Finally, the suspension underwent dialysis against purified water utilizing 3 kDa dialysis membranes for 2 days, with water changes performed every 3 h. Finally, the sample was freeze-dried in preparation for in vitro human fecal fermentation.

#### 2.4.2. In Vitro Fecal Fermentation

After the in vitro upper gastrointestinal tract digestion, the residue was applied to investigate the intestinal probiotic properties [18]. After informed consent was obtained from the health donor, the fecal sample specimens were collected. A sufficient fresh mid-section fecal sample was taken and divided into a fecal collection box, transferred to an anaerobic workstation (Bactron300-2, Shellab Bactron company, Cornelius, NC, USA), and then mixed with PBS buffer (basic phosphate-buffered saline with P1 meta solution) in a 3:1 (buffer volume to fecal weight) to make a fecal slurry. The slurry was filtered through four layers of gauze into a beaker. Then, 1 mL of fecal suspension was mixed with 4 mL of PBS medium containing 50 mg each of residue. Then, it was incubated at 37 °C and 120 rpm. Gas accumulation was measured with a syringe at 0, 6, 12, 24, and 48 h, then centrifuged at 13,000 rpm for 5 min. The supernatants were then stored at −80 °C for SCFAs and pH value text, and the precipitates were used for DNA extraction and sequence.

#### 2.4.3. SCFAs Analysis

SCFAs, including acetate, propionate, and butyrate, were quantified following the method in previous research conducted with slight modifications [19,20]. First, 400 μL of the sample was mixed with 80 μL of HCl (1 mol/L) and 65 μL of 2-ethylbutanoic acid (10 mmol/L) after mixing by vortexing. Then, 1 μL of this mixture was injected into a gas chromatograph (7080B, Agilent, Santa Clara, CA, USA), which was fitted with a capillary column (HP-FFAP, 30 m × 0.25 mm × 0.25 μm, Agilent, Santa Clara, CA, USA). The carrier gases, nitrogen, hydrogen, and air, flowed at rates of 1.5, 40, and 400 mL/min, respectively. The injector and detector temperatures were set at 200 °C and 240 °C, respectively. The temperature program started at 60 °C, increased to 160 °C at a rate of 10 °C/min and maintained for 1 min, raised to 200 °C at the same rate, and finally remained steady at 220 °C for an additional 5 min.

#### 2.4.4. The 16S rRNA Sequencing and Bioinformatics

Total DNA from fermentation precipitates was extracted according to the method described by Luo et al. [21], with molecular size confirmed by 0.8% agarose gel electrophoresis and quality assessed via UV spectrophotometry. PCR amplified the V3–V4 region of the 16s rRNA gene using specific primers 338F (5′-barcode+ACTCCTACGGGAGGCAGCA-3′) and 806R (5′-GGACTACHVGGGTWTCTAAT-3′). The PCR product was purified with an Axygen kit (Corning, NY, USA), quantified by PicoGreen (Invitrogen, Carlsbad, CA, USA), and sequenced on Illumina MiSeq platform with MiSeq Reagent Kit v3 after library preparation. The DADA2 pipeline within QIIME2 was utilized to refine the fastq sequences [22]. Initially, raw data were demultiplexed and primers trimmed using the demux and cutadapt plugins, respectively [23]. Subsequently, sequences underwent quality control, error correction, and chimera detection, culminating in an Amplicon Sequence Variant (ASV) abundance table [22]. Non-singleton ASVs were aligned with mafft and a phylogenetic tree was constructed using fasttree2 [24,25]. Diversity indices and distances were calculated, and taxonomy was assigned to ASVs using the Klearn naive Bayes classifier with the Greengenes database for reference [26,27]. Those analyses were conducted by Shanghai Personalbio Technology Co., Ltd. (Shanghai, China).

### 2.5. Statistical Analysis

All experiments were conducted in triplicate and the results are presented as the mean ± standard deviation unless otherwise stated. Statistical analyses were performed using SPSS Statistics software (version 26.0, Ine. Chicago, IL, USA) and R 4.1.3. Differences in gas production, pH value, and SCFAs production between groups were analyzed using one-way analysis of variance (ANOVA) followed by Tukey’s post hoc test. A significance level of *p* < 0.05 was considered statistically significant for all tests. The α-diversity was determined using the sparse curves on the Shannon index. A principal coordinate analysis (PCoA) plot was performed using the weighted UniFrac distance between bacterial community structures.

## 3. Results and Discussion

### 3.1. In Vitro Digestibility of Starch in Noodles

The digestibility of starch in noodles was assessed in vitro, revealing that germination elevated the digestibility of brown rice noodles (Figure 1, *p* < 0.05). Accordingly, compared to the BRN, the RDS content of GBRN was increased (Table 1). The SDS contents in BRN and GBRN were not significant. However, the RS content was increased in brown rice noodles after germination. Meanwhile, the C_∞_ and k of the digestion also increased in brown rice noodles after germination; these were estimated using the first-order equation model. These values indicate that the amount and rate of digestibility increases as digestion progresses, suggesting that germination improves the digestibility of brown rice noodles. This may be attributed to germination-induced changes in the chemical composition and physical properties of starch or brown rice noodles. Our previous research indicated an increase in both the free and bound phenolic acid content of brown rice after germination [9]. Polyphenols and phytic acid have an inhibitory effect on starch digestion [28]. However, during germination, endogenous enzymes are activated, which causes the dissociation of complexes of polyphenols with dietary fiber, starch, and other components, as well as the decomposition of phytic acid [28]. Therefore, the contact area of polyphenols of germinated brown rice noodles with boiling water increases and the rice is more easily destroyed, leading to a higher digestibility for germinated brown rice noodles. Furthermore, the smooth and densely packed starch granules of brown rice are transformed into rough and eroded shapes during germination, facilitating easier enzymatic degradation [29].

### 3.2. In Vitro Fermentation of Noodles

#### 3.2.1. Gas Production and pH Value

Gas production is a general indicator that reflects the fermentability of available carbohydrate. At each fermentation time point, the gas production of BRN, GBRN, and FOS was significantly higher compared to the Blank group (Figure 2A, *p* < 0.05), indicating that the gut microbiota had a certain fermentation utilization for GBRN, BRN, and FOS. Gas accumulation curves for BRN and GBRN exhibited a two-phase fermentation pattern. At 6 h of fermentation, the gas accumulation of GBRN was significantly less than that of BRN (*p* < 0.05). The content of bound phenolic acid in GBRN was higher than BRN [9]. The bound phenolic acid could inhibit the fermentation of substrate by gut microbiota, resulting in lower gas accumulation in GBRN compared to that in RBN [30]. There was no significant difference between GBRN and BRN after 6 h of fermentation (*p* > 0.05). This indicated that the degree of BRN fermentation was higher than GBRN in the early stage of fermentation.

During fermentation, intestinal flora produce SCFAs, leading to a decrease in the pH of the fermentation liquid. An acidic environment can promote the growth of beneficial bacteria and inhibit the growth of pathogenic bacteria [31]. The pH results of each experimental group are depicted in Figure 2B. The pH values of BRN and GBRN at each fermentation time point are significantly lower than those of the Blank group and higher than those of the FOS group (*p* < 0.05). From 0 to 6 h, the pH of BRN and GBRN decreased rapidly; after 6 h, the pH value of each experimental group gradually stabilized. At 48 h, the pH value of the BRN group was 4.82, which was significantly lower than that of the GBRN group, with a value of 4.96 (*p* < 0.05). This may be because during the initial stages of fermentation, intestinal flora can readily absorb and utilize nutrients to proliferate rapidly and produce SCFAs, thereby reducing the pH value. As the fermentation time increases and the nutrients decrease, the ability of the intestinal flora to produce SCFAs decreases, and the change in pH value is not significant. The above results indicate that germination did not significantly affect gas production but increased the pH of the fermentation liquid during the fermentation process of brown rice noodles, but both BRN and GBRN showed beneficial intestinal fermentation characteristics.

#### 3.2.2. SCFAs Production

To better understand the impact of germination on the fermentation characteristics of brown rice noodles, the production of SCFAs was investigated, as shown in Figure 3. As the fermentation time increased, the production of total SCFAs, acetic acid, propionic acid, and butyric acid also changed continuously among the experimental groups, indicating that the growth status of the intestinal flora was inconsistent at different fermentation periods of the substrate. The production of SCFAs (including total SCFAs, acetate, propionate, and butyrate) of BRN and GBRN was greater than that in the Blank, indicating that BRN and GBRN were all beneficial to intestinal health. The acetate and butyrate production of BRN were greater than that of GBRN (Figure 3A,C). However, the propionate production of BRN was less than that of GBRN (Figure 3B). The total SCFAs production of BRN was larger than GBRN during 6–24 h of fermentation, but exhibited no significant differences at 48 h of fermentation (Figure 3D). The SCFAs distribution profiles provide insight into the microbial metabolism of brown rice noodles. The synthesis pathways of acetic acid and butyric acid are partially similar, while propionic acid has a separate synthesis pathway [32]. Meanwhile, the content of RS in GBRN was lower than that of BRN, and the RS usually increased the production of acetic acid and butyric acid [33]. The γ-aminobutyric acid was reported to promote the production of propionate [14]. The growth of gut microbiota also is an important factor influencing the production of SCFAs. These factors collectively contribute to the variations observed in SCFAs between GBRN and BRN. The above results show that germination could slightly influence the SCFA production during the fermentation of brown rice noodles. In addition, BRN and GBRN also have certain probiotic properties in the intestine.

### 3.3. The Diversity of Gut Microbiota after Noodles Fermented

The α diversity, as measured by the Shannon index, showed dynamic changes over the fermentation time course as well. Figure 4A shows that this sequencing has basically covered all the microorganisms in the intestinal tract after noodles fermented. At 48 h of fermentation, compared to the initial value, the richness gradually increased in the Blank group. BGN and GBRN showed a lower Shannon index than that of Blank. Furthermore, the Shannon index of GBRN was lower than that of BRN. Therefore, the diversity of gut microbiota was increased with the increase in the time of fermentation. The brown rice noodles could decrease the microbiota diversity. Meanwhile, the germination could enhance the effect of decreasing diversity. This may be due to the difference in the structure of brown rice after germination.

The β diversity was measured by the PCoA plot, based on weighted UniFrac distances, and a consistent pattern of microbial community shifts was revealed across different groups (Figure 4B). PCoA primarily occurred along the first principal coordinate (PCo 1), which explains 69.8% of the variation. On the PCo 1 axis, separation was observed between the Blank and the Initial sample, indicating that fermentation could significantly change the microbiota structure. A large distance existed between the noodles group and the Blank group. Meanwhile, the distance between the BRN and GBRN was also large, indicating a substantial shift in the gut microbiota composition. Therefore, the β diversity of gut microbiota was markedly altered by the incorporation of both BRN and GBRN.

### 3.4. Microbiota Adaptation to Brown Rice Noodles

To better compare the differences in the intestinal fermentation microbial structure of BRN and GBRN, the relative abundance of the phylum and genus level of the intestinal flora at 48 h of fermentation are shown in Figure 5. Initially, the fermentation process was predominantly inhabited by the phylum Firmicutes, Bacteroidetes, Actinobacteria, Proteobacteria, and Fusobacteria. After 48 h of fermentation, the relative abundance of Firmicutes and Proteobacteria decreased, and the relative abundance of Bacteroidetes, Actinobacteria, and Fusobacteria increased in the Blank group (Figure 5A). Compared to the Blank group, BRN markedly elevated the relative abundance of Firmicutes, Actinobacteria, and Proteobacteria, but the relative abundance of Bacteroidetes decreased. Furthermore, the relative abundance of Firmicutes in GBRN was higher, and the Bacteroidetes was lower compared to BRN. An elevated presence of Firmicutes coupled with a diminished presence of Bacteroides facilitates the decomposition of carbohydrates [34,35]. Therefore, the germination could promote the decomposition of carbohydrate of brown rice noodles in the host gut. Acidic environments favor the proliferation of butyrate-producing Firmicutes and concurrently impede the expansion of acid-sensitive Bacteroidetes [36], aligning with the observed pH trends in our experiments.

The relative abundance of the genus level composition of the intestinal flora is shown in Figure 5B. The bacteria were mainly composed of *Megamonas*, *Bacteroides*, and *Bifidobacterium* during the initial fermentation. After 48 h, the relative abundance of *Megamonas* decreased, while the *Bacteroides* in the Blank group increased compared to that in the Initial group. However, in the BRN group, the relative abundance of both *Megamonas* and *Bifidobacterium* increased, whereas *Bacteroides* decreased compared to that of the Blank group. Following the germination of brown rice, in the GBRN group, the relative abundance of *Megamonas* increased, but there was a decrease in both *Bacteroides* and *Bifidobacterium* compared to that in the BRN group. The RS of BRN was higher than that of GBRN, and the RS of brown rice noodles belonged to the RS3 (a starch structure formed by retrogradation after gelatinization), and it was reported that it could increase the relative abundance of *Bacteroides* and *Bifidobacterium* [37]. *Megamonas* is a representative bacterium in the intestine that can ferment various carbohydrates to produce acetic acid, propionic acid, and lactic acid [38]. The growth of *Bacteroides* and *Bifidobacterium* is recognized for its contribution to the biosynthesis of acetate and butyrate [32]. Therefore, the germinated could decrease the production of acetate and butyrate after noodles fermented. Additionally, during the germination, its endogenous enzymes are activated, the dissociation of complexes of polyphenols with dietary fiber, starch, and other components [28]. There may be fewer dietary fiber and polyphenols of GBRN than BRN, and these are beneficial to human health [39]. The content of dietary fiber and polyphenols could influence the growth of gut microbiota. 

## 4. Conclusions

This study showed that germination changed the starch digestion characteristics and the intestinal fermentation characteristics of brown rice used to make noodles. Germination promoted the digestion of starch in brown rice noodles, increased the RDS content, and reduced the RS content. This outcome is potentially due to germination disrupting the integrity of brown rice’s cellular matrix and starch granules, thereby enhancing the conversion of RDS to RS. Germination promoted the production of propionate after the brown rice noodles were fermented by intestinal flora while inhibiting the fermentation rate and the production of acetate and butyrate. Furthermore, germination reduced the diversity of the flora after rice noodles were fermented and promoted the growth of *Megamonas* but inhibited the growth of *Bacteroides* and *Bifidobacterium*. These were maybe relative to changes in the bound phenolic acid, RS, dietary fiber, and γ-aminobutyric acid. Overall, germination can improve the nutritional properties of brown rice noodles, and this work provides an interesting idea for functional rice noodles. Nevertheless, the elevated RDS levels in sprouted brown rice noodles could potentially cause a swift surge in postprandial blood glucose, posing health risks for individuals with diabetes. At the same time, differences in microbiota structure between different populations and differences between types of brown rice may lead to different probiotic properties of sprouted brown rice noodles. Therefore, further investigation is needed to explore these effects across various populations (including the chronic disease patients) and brown rice types to better understand the broader implications for human nutrition and food science.

## Figures and Tables

**Figure 1 foods-13-02279-f001:**
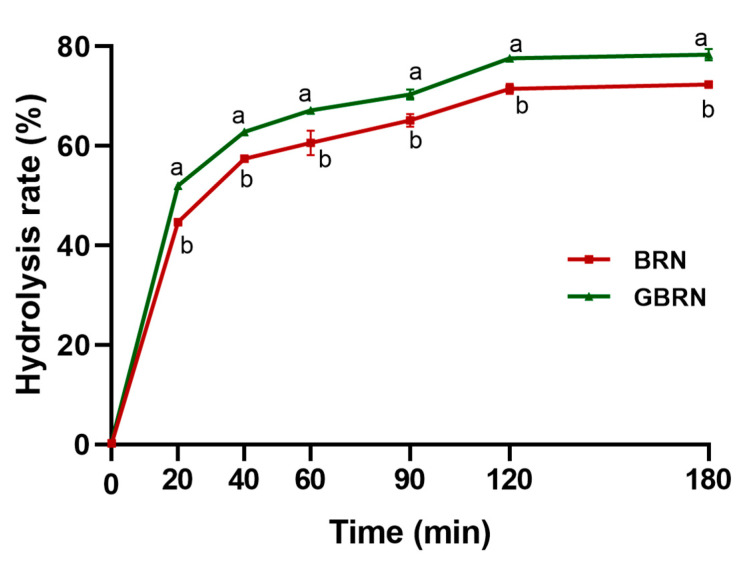
*In vitro* digestion curves of starch in brown rice noodle (BRN) and germinated brown rice noodle (GBRN). At the same time point, different letters represent significant differences between groups.

**Figure 2 foods-13-02279-f002:**
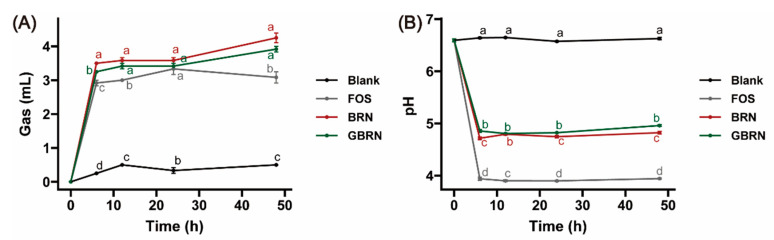
Fermentation profile of BRN and GBRN. (**A**) Gas production. (**B**) pH value. At the same time point, different letters represent significant differences between groups.

**Figure 3 foods-13-02279-f003:**
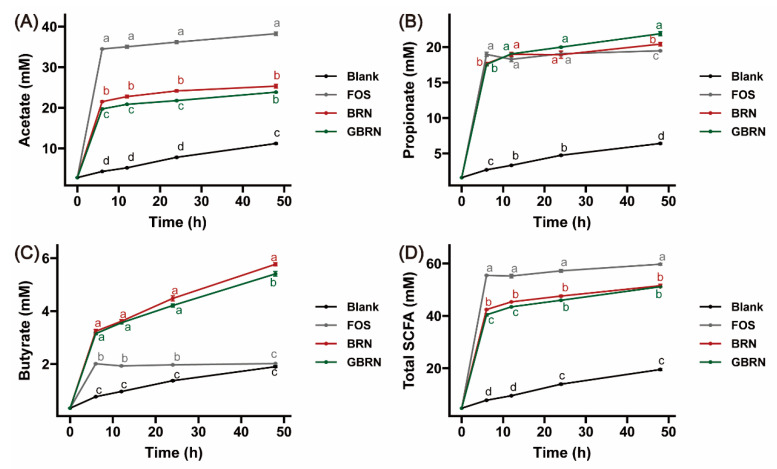
Short-chain fatty acid of BRN and GBRN. (**A**) Acetate. (**B**) Propionate. (**C**) Butyrate. (**D**) Total SCFAs production. At the same time point, different letters represent significant differences between groups.

**Figure 4 foods-13-02279-f004:**
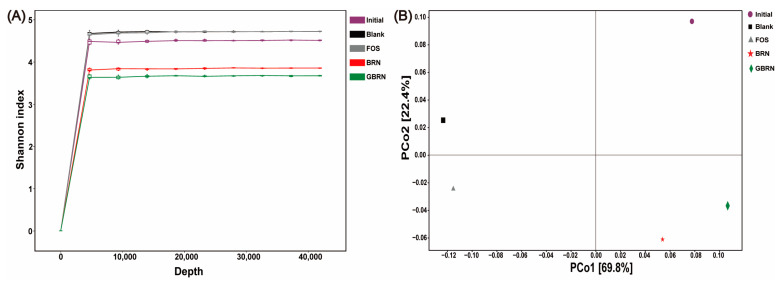
Impact of germination on microbial community diversity during in vitro fecal fermentation of brown rice noodles. (**A**) Sparse curves: Shannon index. (**B**) PCoA based on weighted UniFrac distances. Initial, Blank, FOS, BRN, and GBRN stand for the initial state of fermentation, blank control, fructo-oligosaccharide group, brown rice noodle group, and germinated brown rice noodle group, respectively.

**Figure 5 foods-13-02279-f005:**
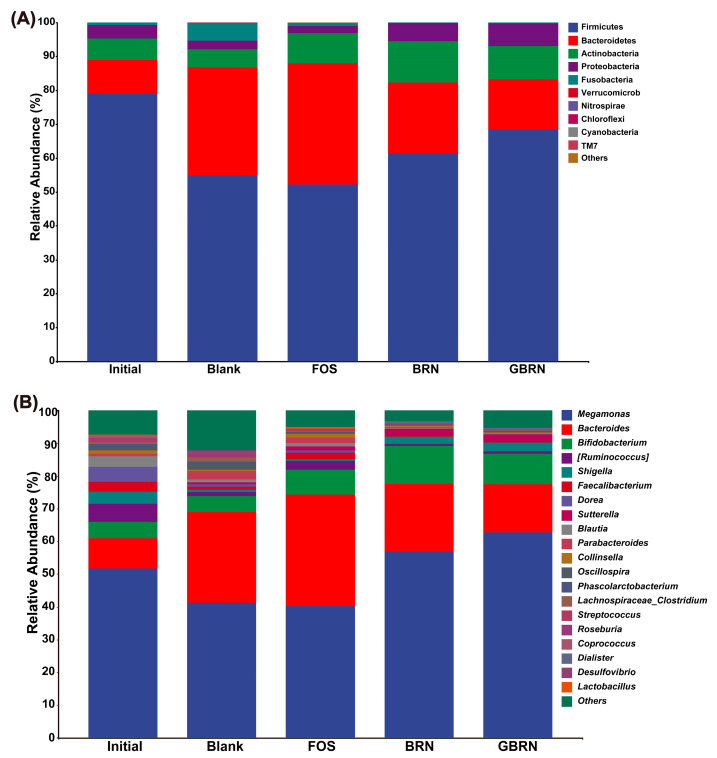
Impact of germination on microbial community composition during in vitro fecal fermentation of brown rice noodles. (**A**) Microbiota taxonomic classification at the phylum level. (**B**) Microbiota taxonomic classification at the genus level. Initial, Blank, FOS, BRN, and GBRN stand for the initial state of fermentation, Blank control, fructo-oligosaccharide group, brown rice noodle group, and germinated brown rice noodle group, respectively.

**Table 1 foods-13-02279-t001:** *In vitro* digestibility parameters of starch in brown rice noodle and germinated brown rice noodle.

	BRN	GBRN
RDS (%)	44.68 ± 0.36	52.06 ± 0.22 *
SDS (%)	27.71 ± 0.20	26.28 ± 1.02
RS (%)	27.61 ± 0.55	21.66 ± 1.14 *
C_∞_ (%)	72.39 ± 0.55	78.34 ± 1.14 *
k × 10^−2^ (min^−1^)	1.44 ± 0.05	1.90 ± 0.04 *

The data were presented as mean ± standard deviation of triplicates. An asterisk (*) indicates a statistically significant difference (*p* < 0.05) between the two groups for the same parameter. BRN stands for brown rice noodle, and GBRN stands for germinated brown rice noodle, respectively. RDS, SDS, RS, C_∞_, and k stand for rapidly digestible starch, slowly digestible starch, resistant starch, the equilibrium hydrolysis percentage, and the first-order rate coefficients of the starch digestion of noodles.

## Data Availability

The original contributions presented in the study are included in the article, further inquiries can be directed to the corresponding author.

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
