# Peer review of "Germination-Induced Enhancement of Brown Rice Noodle Nutritional Profile and Gut Microbiota Modulation"

_foods, 2024, doi:10.3390/foods13142279_

Round 1

Reviewer 1 Report

Comments and Suggestions for Authors

The objective of the work was to investigate the digestion characteristics of starch and the intestinal probiotic properties of brown rice noodles and germinated brown rice noodles by in vitro. The work provides evidence of the fermentation process of starch in vitro, which shows the effect of the germination process.

General concept comments

Include more information about the reported mechanisms of the process of germination of food and their effects on nutrition.

Title: Germination-induced enhancement of brown rice noodle nutritional profile and gut microbiota modulation

Abstract: Rewrite this section in chronological order and include a brief description of the methodology used.

Keywords: No comments

Introduction: Complement with information about the mechanisms of the fermentation process (endogenous enzymes, the dissociation of complexes of polyphenols with dietary fiber, starch, and other components) and the effects on the foodstuff subjected to the process, as well as the effect on the intestinal microbiota.

Materials and methods: No comments

Results and discussion:

In the case of the results presented for RDS and RS, both variables could be related, i.e. that the germination process allowed the liberation of part of the RS starch, becoming RDS. So, by having a greater amount of RDS, the fermentation is promoted, preferably by the propionate producing path, which could be included in the discussion. Besides, the described bacteria genera are also favored.

The discussion could be more comprehensive and include all or most of the variables analyzed.

The graph about the bacterial diversity shows just a few genera, could you explain why?

It would be desirable to complement the information with the analysis of the chemical composition of the brown rice noodles, germinated and non-germinated, to identify the differences in fiber content.

The greater effect is observed with a higher availability of starch. However, there is no evidence of what happened to the fiber or the phenols, which were mentioned previously.

Conclusion: Please include the limitations of the present study.

References: No comments

Please italicize”in vitro” in the whole document.

Reviewer 2 Report

Comments and Suggestions for Authors

The authors sought to reproduce in detail the conditions prevailing during the digestion of digested brown rice products, reflecting on the effects of germination on assimilation and the gut microbiota.

The question is interesting from a food safety and public health point of view, due to the popularity of this type of food worldwide.

The topic is original.

This  is a subject that has been little discussed and is therefore innovative.

The article is written correctly, but Figure 2a is not described in the text of the manuscript. 

The text is clear and easy to read.

 The conclusions are too general and do not reflect the analysis presented in the article. The paper is very interesting and the results are well elaborated. The ideal conditions in the study e.g. distilled water eliminate e.g. the influence of ions found in drinking water, this is worth considering. The authors report a decrease in butyrate production, and as is well known, butyrate is an energy source for colonocytes, influences the regeneration and accelerated healing of the intestinal epithelium, supports intestinal barrier function, has anti-inflammatory properties and improves peristalsis of the large intestine. so will the decrease be beneficial ? the composition of the intestinal microbiota is a very individual characteristic, so the effect of functional foods made from sprouted rice should be further analysed - on whom it may have a negative effect. 

Round 2

Reviewer 1 Report

Comments and Suggestions for Authors

Title: Germination-induced enhancement of brown rice noodle nutritional profile and gut microbiota modulation.

The objective of the work was to investigate the digestion characteristics of starch and the intestinal probiotic properties of brown rice noodles and germinated brown rice noodles. The work provides sufficient evidence of the fermentation process of starch in vitro, which shows the effect of the germination process.

The document was substantially improved by the inclusion of additional information.

General comments

Abstract: No comments

Keywords: No comments

Introduction: No comments

Materials and methods: No comments

Results and discussion:

Lines 178-179: The abbreviations BRN and GBRN in Table 1 can be taken out from the title and added, instead, to the description beneath it.

Lines 180-183: Reduce the space between rows in Table 1.

Table 1: Reduce its size.

Line 200: Please give the difference (in number or percentage) of the pH value. The phrase “less acid” might be more suitable than “greater”.

Line 208: In the description of Figure 2, does Gas Production refer to the amount of accumulated gas per time?

Line 217: Please place a comma between the letters in (Figure 3AC) because it can be interpreted that it refers to figures from A to C.

Lines 218-219, 224-226: Check which abbreviation, SCFAs or SCFA, is correct and homogenize.

Lines 248-249, 279, Figures 4 and 5: Add the meaning of the abbreviations used in the figures to the description.

Lines 265-267: Please indicate which treatments are being compared in the following segment: “After 48 h, the relative 265 abundance of Megamonas decreased, and Bacteroides increased. However, compared to Blank group, the relative 266 abundance of Megamonas and Bifidobacterium increased, but Bacteroides decreased. After brown rice germinated, the 267 relative of abundance of Megamonas increased, but Bacteroides and Bifidobacterium decreased”.

Lines 269-270: Check the font type and size in “(a starch structure formed by retrogradation after gelatinization) which was reported that could increase the relative abundance of Bacteroides, and Bifidobacterium”.

Conclusion: No comments

References: No comments
